# Direct Evidence of Powassan Virus Vertical Transmission in *Ixodes scapularis* in Nature

**DOI:** 10.3390/v16030456

**Published:** 2024-03-16

**Authors:** Rachel E. Lange, Melissa A. Prusinski, Alan P. Dupuis, Alexander T. Ciota

**Affiliations:** 1Department of Biomedical Sciences, School of Public Health, State University of New York, Albany, NY 12144, USA; rlange@albany.edu (R.E.L.);; 2The Arbovirus Laboratory, New York State Department of Health, Wadsworth Center, Slingerlands, NY 12159, USA; 3Vector Ecology Laboratory, New York State Department of Health, Bureau of Communicable Disease Control, Albany, NY 12237, USA; melissa.prusinski@health.ny.gov

**Keywords:** Powassan virus, deer tick virus, *Ixodes scapularis*, blacklegged tick, arbovirus, zoonosis, vertical transmission, New York State

## Abstract

Powassan virus (POWV) is a tick-borne flavivirus endemic in North America and Russia. Experimental infections with POWV have confirmed horizontal, transstadial, vertical, and cofeeding transmission routes for potential virus maintenance. In the field, vertical transmission has never been observed. During New York State tick-borne pathogen surveillance, POWV RNA and/or infectious POWV was detected in five pools of questing *Ixodes scapularis* larvae. Additionally, engorged female *I. scapularis* adults were collected from hunter-harvested white-tailed deer (*Odocoileus virginianus*) in a region with relatively high tick infection rates of POWV and allowed to oviposit under laboratory conditions. POWV RNA was detected in three female adult husks and one pool of larvae from a positive female. Infectious virus was isolated from all three RNA-positive females and the single positive larval pool. The detection of RNA and infectious virus in unfed questing larvae from the field and larvae from replete females collected from the primary tick host implicates vertical transmission as a potential mechanism for the maintenance of POWV in *I. scapularis* in nature, and elucidates the potential epidemiological significance of larval ticks in the transmission of POWV to humans.

## 1. Introduction

Powassan virus (POWV; family *Flaviviridae*) is a flavivirus maintained by ticks belonging to the family Ixodidae in North America and Russia (reviewed in [1,2]). The first human case was detected in Canada in 1958 followed by the first reported human case in the United States (USA) in 1970 [3,4]. Since its initial identification, POWV has been identified in various tick genera and in human case reports primarily in the Northeastern and Midwestern regions of the USA [5]. Common clinical manifestations of POWV during human infection include fever, weakness, and nausea during acute infection followed by encephalitis, neurological deficits, and paralysis during neuroinvasive infection [1,2,5]. Mortality rates from POWV infection range from 10 to 12% with 50% of neuroinvasive disease survivors displaying lifelong neurological issues [1,2,5]. There has been a notable increase in POWV human cases, POWV-infected ticks, and serological evidence in mammals in recent decades suggesting increases in both geographic range and prevalence [5].

POWV comprises two genetically distinct lineages: Powassan virus, lineage 1 (POWV-1), and deer tick virus (DTV), lineage 2. These lineages are maintained in ecologically discrete enzootic cycles (reviewed in [1,6]). Lineage 1 is associated with the primarily nidicolous tick species, *Ixodes cookei* and *I. marxi,* and their hosts, groundhogs, mustelids, and arboreal squirrels [7,8]. Lineage 2 is primarily detected in the more ubiquitous *I. scapularis* [9]. There is no confirmed vertebrate reservoir for POWV, but recent evidence suggests shrews may play a role in DTV maintenance [1,10]. Recent reports have identified alternative tick species including *Dermacentor* spp., *Haemaphysalis longicornis*, and *Amblyomma americanum* as competent vectors for DTV with maintenance confirmed through horizontal and transstadial transmission [11,12]. The significance of these alternative host species in POWV maintenance in nature remains unknown.

POWV maintenance within tick hosts has only been explored through experimental infection of various competent tick species. Costero and Grayson found that POWV is horizontally, transstadially, and vertically transmitted in experimentally infected *I. scapularis* [13]. The transmission of POWV through these mechanisms has also been reported in *D. andersoni*, *D. variabilis*, *A. americanum*, and *H. longicornis* [11,12,14]. Few studies have been conducted to understand the relative importance of these modes of transmission in nature. However, the low incidence and focal nature of POWV have implicated vertical transmission and cofeeding as potential primary mechanisms for maintenance [15,16]. While vertical transmission rates are typically low in tick-borne viruses (<10%), it is known to play an important role in the ecological maintenance of tick-borne encephalitis virus (TBEV), Omsk hemorrhagic fever virus (OHFV), and Kyasanur Forest disease virus (KFDV) [17,18,19]. Through our unique access to diverse tick species, life stages, and regional populations with reported POWV incidence, we were able to detect and confirm the first known evidence of vertical transmission of POWV in *I. scapularis* collected in nature in New York State.

## 2. Materials and Methods

### 2.1. Larval Tick Collections and Testing

The New York State Department of Health (NYSDOH) has been conducting statewide tick surveillance on public lands in NYS for POWV since 2008. Questing ticks were collected by standard drag and flag techniques as previously described [20]. Briefly, a 1 m^2^ piece of white cloth was dragged over low vegetation and leaf litter to obtain host-seeking ticks. Ticks were removed from the cloth and kept alive at 4 °C and 100% humidity until species, developmental stage, and sex identification. Ticks were pooled and fresh-frozen at −80 °C for testing. All pools were tested for POWV RNA by an in-house multiplex quantitative reverse transcriptase polymerase chain reaction (qRT-PCR) assay targeting POWV nonstructural protein 5 as previously described [20]. Briefly, larval pools were homogenized in 0.6 mL of diluent (20% heat-inactivated FBS in Dulbecco PBS, 50 µg/mL penicillin/streptomycin, 50 µg/mL gentamicin, and 2 µg/mL fungizone [Sigma-Aldrich, St. Louis, MO, USA]) using a Retsch Mixer Mill, MM 301 (Retsch Inc., Newtown, PA, USA) at 30 cycles/second for a 4 min cycle. RNA was extracted using an automated MagMAX nucleic acid extraction kit and associated instrument (ThermoFisher Scientific, Waltham, MA, USA), and the extracted RNA was used in the multiplex POWV qRT-PCR assay.

### 2.2. Replete Female Collection, Maintenance, and Larval Testing

Adult *I. scapularis* were collected from hunter-harvested white-tailed deer (*Odocoileus virginianus*) brought to private deer processing stations in Saratoga County in November of 2022. Deer were examined for 15 min for *I. scapularis* and blood samples were collected from the carcass, as described previously [20]. Engorged adult female *I. scapularis* were stored alive at room temperature in 1.5 mL conical tubes (ThermoFischer Scientific, Waltham, MA, USA). Deer blood was stored in 1.5 mL conical tubes and held at 4 °C until processing.

Fully replete *I. scapularis* were transferred to individual 7 mL round bottom cell culture tubes (ThermoFischer Scientific, Waltham, MA, USA) and maintained in the laboratory at 95% relative humidity (RH), 20 °C, and on a 16:8 light/dark cycle. Females were held for 6 months to allow for oviposition and larval hatching. Female husks were collected from those that laid egg batches and tested for POWV RNA by multiplex POWV qRT-PCR assay described above. For each egg batch that produced viable larvae, half of the larvae were collected, pooled, and flash-frozen at −80 °C for POWV RNA testing.

### 2.3. Hunter-Harvested White-Tailed Deer Blood Testing

Deer blood collected from hunter-harvested white-tailed deer in Saratoga County, NY, was used to determine POWV seropositivity by plaque reduction neutralization testing (PRNT) as previously described [20]. Briefly, deer blood was tested against prototype POWV (LB strain) with a 90% cut-off at serum dilutions of 1:10.

### 2.4. Cells and Live Virus Confirmation

Baby hamster kidney cells (BHK-21, ATCC, CCL-10) were grown in minimal essential media (MEM, Gibco, Invitrogen Corp, Carlsbad, CA, USA) supplemented with 10% heat-inactivated fetal bovine serum (FBS, Hyclone, Logan, UT, USA). Cells were maintained at 37 °C, 5% CO_2_. For viral confirmation, T25 cell culture flasks (ThermoFischer Scientific, Waltham, MA, USA) were seeded and allowed to grow to confluency. Confluent flasks were inoculated with POWV RNA positive tick homogenates and allowed to incubate for 1 h at 37 °C, 5% CO_2_. After the incubation, 5 mL of maintenance media was added, and cultures were monitored for 96 h post-infection (HPI) for cytopathic effects (CPE). The supernatant was subsequently harvested at 24 h intervals and tested by multiplex POWV qRT-PCR assay for POWV RNA confirmation. Additionally, culture supernatant was tested for live virus by plaque assay as previously described [21]. Briefly, confluent six-well cell culture plates were inoculated in duplicate and allowed to incubate for 1 h before the addition of a 3 mL semisolid overlay of 0.6% oxoid agar and maintenance media. A second overlay of 0.6% oxoid agar, maintenance media, and 2% neutral red was added 3 days post-inoculation and plaques were counted the following day. Additionally, culture supernatant from 96 DPI was harvested, filtered, and passed on to fresh confluent BHK-21 cultures where CPE was further monitored.

## 3. Results

### 3.1. POWV-Positive Questing I. scapularis Larvae Detection in NYS

Questing tick surveillance has been conducted statewide on public lands in NYS for tick-borne viruses since 2008. Using standardized flagging and dragging methods, *I. scapularis* of all life stages were frequently collected and tested for POWV RNA using a qRT-PCR assay. From 2017 to 2023, 30,580 *I. scapularis* larvae were collected and tested from 46 of 62 NYS counties (2017 *n* = 4779, 2018 *n* = 4223, 2019 *n* = 5178, 2020 *n* = 3931, 2021 *n* = 3772, 2022 *n* = 1875, and 2023 *n* = 6822). All *I. scapularis* larvae were collected during 644 individual sampling events from 223 unique locations. Generally, each location was visited twice per year, and sampling which yielded *I. scapularis* larvae was conducted during the months of May-August. Of the 46 counties sampled, the most larvae were collected from Saratoga (*n* = 5496), Greene (*n* = 2823), Albany (*n* = 2672), Schenectady (*n* = 2361), Schoharie (*n* = 2380), and Rensselaer (*n* = 2126) counties. Over 1000 individuals were collected from five additional counties (Columbia, Montgomery, Oneida, Otsego, and Warren counties). All other counties accounted for less than 1000 larvae collected over the 7-year timeframe. Of the 702 pools tested, 5 pools were positive for POWV RNA (minimum infection rate [MIR]: 0.19%). The five positive pools were detected in 2018 (*n* = 2), 2020 (*n* = 1), and 2023 (*n* = 2) from counties with a history of POWV transmission (1 positive pool each from Greene, Montgomery, Saratoga, Schenectady, and Schoharie counties) (Figure 1).

### 3.2. Vertical Transmission of POWV to Larvae from Replete Adult Female I. scapularis Collected from Hunter-Harvested White-Tailed Deer in NYS

In November 2022, we collected engorged female *I. scapularis* adults from hunter-harvested white-tailed deer from Saratoga County, a county in NYS where POWV is considered endemic, with a consistent detection of POWV infection in host-seeking ticks (~1–5%), high rates of POWV seropositivity in white-tailed deer (~80% from 2007 to 2019), and locally documented acquired human cases of Powassan virus infection (0.07 incidence rate from 2004 to 2022) [21,22]. Engorged females were individually housed and maintained in the laboratory to allow for ovipositing. Egg masses from seven replete females hatched, and the husks from those females were subsequently tested for POWV. Of the seven females, two individuals were collected from deer 17-008, four individuals were collected from deer 44-004, and one individual was collected from deer 20-005 (Table 1). Three females tested positive for POWV RNA, each originating from a different deer, with cyclic threshold (CT) values ranging from 22 to 23 (Table 1). Larvae from the POWV-positive females were counted individually, pooled, and tested. A total of ~1050 individuals were tested from the positive female pulled from deer 17-008 (*n* = 5 pools each consisting of 100–300 larvae), ~800 individuals from the positive female pulled from deer 44-004 (*n* = 4 pools of 200 larvae), and ~1000 individuals from the positive female pulled from deer 20-005 (*n* = 4 pools of 100–300 larvae). One pool (*n* = 300) from the positive female collected from deer 20-005 was positive for POWV RNA with a CT of 38.934 (Table 1).

To confirm these results, the POWV-positive female husk and larval pool from deer 20-005 were tested for infectious virus on cell culture. Female husk and tick pool homogenates were used to inoculate BHK-21 cultures and monitored for 96 HPI. Cultures were checked every 24 h and CPE was recorded. Additionally, culture supernatant was collected at 24 h intervals and tested for live POWV by plaque assay. All cultures displayed CPE by 48 HPI, and the plaque assay revealed virus presence at all timepoints. At 96 HPI, the supernatant was harvested, filtered, and passed on to fresh BHK-21 cultures where CPE was further monitored. CPE was detected by 24 HPI from all passaged cultures.

## 4. Discussion

The results from this study provide the first evidence of vertical transmission of POWV in *I. scapularis* in nature. Previous work has highlighted the roles of horizontal, transstadial, vertical, and cofeeding mechanisms for POWV maintenance, but are limited to experimental studies [13,14]. Experimental infections have used microinjection, immersion, or feeding on laboratory mice to study these modes of transmission, creating a gap in our understanding of POWV transmission mechanisms within natural cycles. Additionally, the lack of a definitive vertebrate reservoir points towards potentially significant roles for these alternative transmission cycles. This is further supported by the focal nature of POWV transmission hotspots, which are dissimilar to the diffuse geographic spread of other tick-borne pathogens like *Borrelia burgdorferi* and *Anaplasma phagocytophilum* [15,23,24,25]. Vertebrate hosts play a dominant role in the transmission and spread of these bacterial tick-borne pathogens, and vertical transmission does not occur or is extremely rare [26,27]. As such, nymphal and adult ticks are considered the developmental stages primarily associated with the transmission of *B. burgdorferi* and *A. phagocytophilum* to humans. The detection of RNA and infectious virus in unfed questing larvae from the field implicates larval *I. scapularis* as potentially significant in POWV ecology and human exposure risk. The need for further understanding the significance of alternative POWV transmission mechanisms is buoyed by the increased detection and study of POWV in non-canonical tick vectors. Recent studies have reported POWV-1 in *I. scapularis* and DTV in *D. variabilis* collected in NYS [21,28]. Furthermore, experimental infections have confirmed vector competence and transmission of DTV in *D. variabilis*, *A. americanum*, and *H. longicornis* [11,12].

While POWV is currently confined to North America and Russia, transmission dynamics of tick-borne flaviviruses endemic to Eurasia may provide insight into POWV maintenance. TBEV is a genetically similar tick-borne flavivirus endemic to an area extending from Europe to eastern Asia. This virus is spread by *I. persculcatus and I. ricinus* with epidemiological and experimental evidence supporting an important role for vertical transmission in viral maintenance in nature [17,29]. Likewise, the vertical transmission plays an important role in the maintenance of OHFV, vectored primarily by *D. reticulatus* in Russia, and KFDV, primarily vectored by *H. spinigera* in India [18,19]. The discovery of natural vertical transmission of POWV in *I. scapularis* and the isolation of POWV from sympatric tick species necessitates further experimental work to assess the potential for POWV maintenance in *I. scapularis* as well as other competent tick genera.

## 5. Conclusions

The detection of vertical transmission of POWV in *I. scapularis* in nature is important in understanding virus dynamics and the natural ecological maintenance and transmission cycles of POWV. Infected host-seeking larvae may also present a POWV exposure risk to people, in contrast with Lyme disease and anaplasmosis where larval ticks do not play a role in pathogen transmission. With increased detection of POWV human cases and expansion of *Ixodes* species in the Northeastern US and Canada within the last two decades, understanding POWV maintenance in nature is key to developing tools to reduce the risk of human infection. Future work should focus on elucidating vertical transmission mechanisms, determining transmission rates, quantifying the epidemiological significance of POWV-infected larval ticks, and assessing the significance of ecological POWV maintenance across its range, all of which will contribute to the development of more effective prevention methods for human populations.

## Figures and Tables

**Figure 1 viruses-16-00456-f001:**
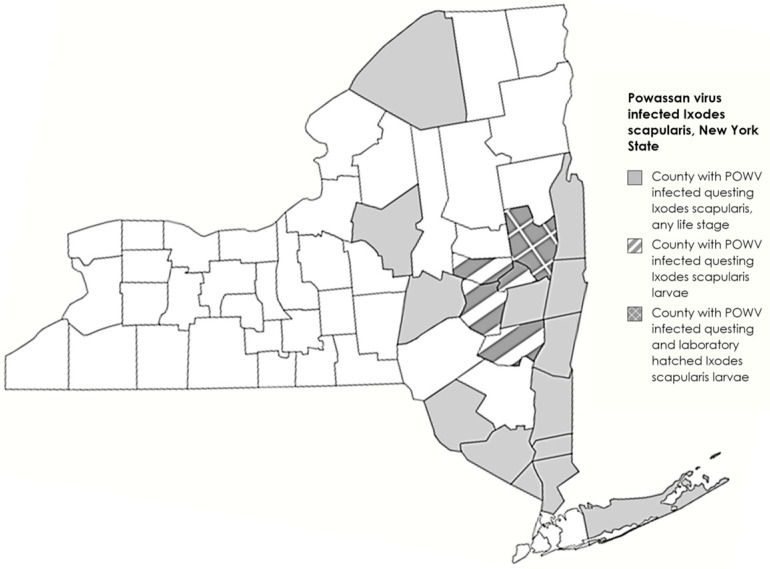
Larval *I. scapularis* collection sites across New York State, USA. Counties in gray represent regions where *I. scapularis* of any life stage have previously been collected with striped counties representing collection of questing *I. scapularis* larvae. POWV-positive questing *I. scapularis* larvae and lab-hatched larvae from engorged female *I. scapularis* were collected in Saratoga County (checkered).

**Table 1 viruses-16-00456-t001:** Powassan virus (POWV) seropositivity in white-tailed deer and virus positivity in ticks from samples collected in Saratoga County, New York State in 2022. All deer were seropositive for POWV (Y = yes), while one engorged *I. scapularis* female and one of its associated larval pools were positive for POWV RNA. POWV RNA was detected by qRT-PCR (cyclic threshold [CT] < 40). Presence of infectious virus (Inf.) was confirmed by cell culture isolation from RNA-positive tick samples (Y = yes, N = no, NA = not applicable). The sample with confirmed vertical transmission is highlighted in **bold**.

			POWV RNA (CT)		Inf. POWV	
Deer	POWV Seropositivity	No. Larvae	Adult (CT)	Larval Pools (CT)	Adult	Larval Pools
22091-17	Y	NA	N	N	NA	NA
22091-17	Y	1050	23.624	N	Y	NA
22091-44	Y	800	23.513	N	Y	NA
22091-44	Y	NA	N	N	NA	NA
22091-44	Y	NA	N	N	NA	NA
22091-44	Y	NA	N	N	NA	NA
**22091-20**	**Y**	**1000**	**22.849**	**38.934**	**Y**	**Y**

## Data Availability

All data are contained within the article.

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
