# Peer review of "Direct Evidence of Powassan Virus Vertical Transmission in Ixodes scapularis in Nature"

_viruses, 2024, doi:10.3390/v16030456_

Round 1

Reviewer 1 Report

Comments and Suggestions for Authors

Manuscript viruses-2905545-peer-review-v.1 presents evidence for vertical transmission of Powassan virus in field populations of the tick Ixodes scapularis, an established vector of the virus.  The results of this field study are straight forward and informative.  They represent an important contribution to present knowledge of the transmission dynamics of Powassan virus.  The methodology is robust and well described.  I have just a few comments.

1. Results, line 126.  “…with high POWV infection rates in host-seeking ticks…”  What do the authors consider to be "high POWV infection" rates?  This statement would be more informative if they add a reference to support this statement?

2. Line 135.  “…n=5 pools of 100-300 larvae…”  It is 5 pools ‘each consisting’ of 100-300 larvae.

Comments on the Quality of English Language

In general the manuscript is clearly written, with minor exceptions.

Reviewer 2 Report

Comments and Suggestions for Authors

Lange te al., proved the vertical transmission of Powassan virus from adult females to larvae of I. scapularis in nature. After laboratory experiments this is the first proof for natural transmission between developmental stages of I. scapularis. The result helps better understanding of natural cycle of this zoonotic antrhopode-borne viral pathogen. The paper is concise, Tables Figures help understanding.

Introduction, lines 30-40. Although the authors cite review articles some sentences should be added here about the present situation and history of POW virus infections in New York state and/or USA. Affected geographical areas, number of human infections and death toll. Spreading? Decreasing? etc.

Are abbreviations like NYS, NYSDOH necessary? Not used too often.

line 118. Were there any association between these positive pools and human cases in these county regions?

lines 122-139.- Only 7 engorged females were collected?

line 134- and three of them were positives? This indicates extremely high incidence of the virus in ticks.

135-137- Did you count the larvae individually?

Reviewer 3 Report

Comments and Suggestions for Authors

This study is straightforward, and the conclusion is clear while the manuscript is well-written. I only have a few comments.

Introduction:

Could you please provide more examples regarding the vertical transmission of tick-borne viruses? How common is vertical transmission?

Additionally, what are the symptoms of the Powassan virus in human?

Line 117: Could you specify how many samples were collected in each county and each year from 2017 to 2023? Also, how frequently were the samples collected?

Reviewer 4 Report

Comments and Suggestions for Authors

Lange R.E. et al: Direct evidence of Powassan virus vertical transmission in Ixodes scapularis in nature

The authors provide a conclusive evidence that Powassan virus is spontaneously transmitted from tick females to their offspring. The article is well written and perfectly fits the scope of Viruses. I recommend it for publication - conditionally on a minor revision explained bellow:

P.2, l.47: substitute “and” for the ampersand, pls. (see Instructions for authors for the Journal’s style)

P.5, l. 183: Eurasia

P.5, l.185: .. endemic to an area extending from Europe to eastern Asia ..

P.5, l.185:    I. persulcatus and I. ricinus

P.4, Table 1:  leave out the first column, pls. – sample codes are not referred to throughout the text, and are thus of no importance to the reader. Sex and age of the sampled animals might be more relevant (if known).  

From p.6, l.226, on: pls., re-format carefully all references in the list according to the Journal’s style (see MDPI Reference List and Citations Style Guide)
